# Differential Antimycotic and Antioxidant Potentials of Chemically Synthesized Zinc-Based Nanoparticles Derived from Different Reducing/Complexing Agents against Pathogenic Fungi of Maize Crop

**DOI:** 10.3390/jof7030223

**Published:** 2021-03-18

**Authors:** Anu Kalia, Jashanpreet Kaur, Manisha Tondey, Pooja Manchanda, Pulkit Bindra, Mousa A. Alghuthaymi, Ashwag Shami, Kamel A. Abd-Elsalam

**Affiliations:** 1Electron Microscopy and Nanoscience Laboratory, Department of Soil Science, College of Agriculture, Punjab Agricultural University, Ludhiana 141004, Punjab, India; 2Department of Microbiology, College of Basic Sciences and Humanities, Punjab Agricultural University, Ludhiana 141004, Punjab, India; grewalj119@gmail.com (J.K.); tondeymanisha@gmail.com (M.T.); 3School of Agricultural Biotechnology, College of Agriculture, Punjab Agricultural University, Ludhiana 141004, Punjab, India; poojamanchanda5@pau.edu; 4Institute of Nanoscience and Technology, Habitat Centre, Phase-10, Sector-64, Mohali 160062, Punjab, India; pulkitbindra207@gmail.com; 5Biology Department, Science and Humanities College, Shaqra University, Alquwayiyah 11726, Saudi Arabia; 6Biology Department, College of Sciences, Princess Nourah bint Abdulrahman University, Riyadh 11617, Saudi Arabia; 7Plant Pathology Research Institute, Agricultural Research Center (ARC), Giza 12619, Egypt; kamelabdelsalam@gmail.com

**Keywords:** metal oxides, nano-fungicides, pathogenic fungi, protein profiling, radical scavenging activity

## Abstract

The present study aimed for the synthesis, characterization, and comparative evaluation of anti-oxidant and anti-fungal potentials of zinc-based nanoparticles (ZnNPs) by using different reducing or organic complexing-capping agents. The synthesized ZnNPs exhibited quasi-spherical to hexagonal shapes with average particle sizes ranging from 8 to 210 nm. The UV-Vis spectroscopy of the prepared ZnNPs showed variation in the appearance of characteristic absorption peak(s) for the various reducing/complexing agents i.e., 210 (NaOH and NaBH_4_), 220 (albumin, and thiourea), 260 and 330 (starch), and 351 nm (cellulose) for wavelengths spanning over 190–800 nm. The FT-IR spectroscopy of the synthesized ZnNPs depicted the functional chemical group diversity. On comparing the antioxidant potential of these ZnNPs, NaOH as reducing agent, (NaOH (RA)) derived ZnNPs presented significantly higher DPPH radical scavenging potential compared to other ZnNPs. The anti-mycotic potential of the ZnNPs as performed through an agar well diffusion assay exhibited variability in the extent of inhibition of the fungal mycelia with maximum inhibition at the highest concentration (40 mg L^−1^). The NaOH (RA)-derived ZnNPs showcased maximum mycelial inhibition compared to other ZnNPs. Further, incubation of the total genomic DNA with the most effective NaOH (RA)-derived ZnNPs led to intercalation or disintegration of the DNA of all the three fungal pathogens of maize with maximum DNA degrading effect on *Macrophomina phaseolina* genomic DNA. This study thus identified that differences in size and surface functionalization with the protein (albumin)/polysaccharides (starch, cellulose) diminishes the anti-oxidant and anti-mycotic potential of the generated ZnNPs. However, the NaOH emerged as the best reducing agent for the generation of uniform nano-scale ZnNPs which possessed comparably greater anti-oxidant and antimycotic activities against the three test maize pathogenic fungal cultures.

## 1. Introduction

Fungal pathogenic diseases are responsible for yield losses in staple calorie and commercial commodity crops posing a major threat to crop productivity globally. The yield gaps have enhanced due to the emergence of new fungal crop pathogens [1] as a consequence of intensive monoculture and environment variations arising due to aberrant climatic conditions [2,3]. Therefore, the agronomic interventions, land management practices, and climate change have been the primary agents that have altered both abiotic and biotic components affecting crop growth and yield [4].

Maize, a versatile cereal-food, feed, and industrial crop, is sensitive to attacks and diseases caused by several fungal pathogens [5]. It is the second-largest considering the area under production and is ranked fourth in productivity among cereals [6] across the globe. Substantial annual yield decreases and depreciation in grain quality in maize are two quantifiable manifestations of the fungal infection and disease [5]. The primary fungal pathogens of maize include the *Macrophomina phaseolina*, *Curvularia lunata* and *Fusarium oxysporum* which cause charcoal rot, leaf spot, and stalk rot diseases respectively in maize. The infected plants are generally treated with anti-fungal compounds or fungicides to curb the spread of the pathogen. However, for continuous monoculture predominated agroecosystems, spraying of these antifungals only provides ephemeral protection to the plants due to the single target site mechanism and the emergence of resistant fungal strains [1]. This necessitates the development of effective analogous antifungals without compromising the ecological and bio-safety aspects.

The present decade has witnessed the emergence and use of nano-scale materials as potent anti-microbials particularly anti-fungal agents. The zinc nanomaterials including the nano-zinc particles possess excellent anti-mycotic properties against a variety of plant fungal pathogens [7,8,9,10,11,12]. The predominant mechanisms governing the anti-mycotic effect of ZnNPs include the reactive oxygen species enabled stress besides Zn^2+^-based toxicity occurring due to the formation of these ions on dissolution of ZnNPs in the cell environment [7]. The size of ZnNPs and their crystal chemistry can affect the fungicidal potential as these characteristics alter the ability to trespass the fungal cell wall and membrane structures to elicit ROS response besides varying the dissolution of the ZnNPs within the fungal cell cytoplasm. The anti-fungal activity of ZnNPs have been identified against *Alternaria alternata* [13], *Aspergillus flavus* [14], *Botrytis cinerea* [11,15], *Candida albicans* [16], *Fusarium graminearum* [17], *Fusarium moniliforme* [8], *Fusarium solani* [18], *Penicillium* sp. [19,20,21], *Penicillium expansum* [22], *Pythium ultimum* and *P. aphanidermatum* [23], *Rhizopus stolonifera* [24] and many more fungal pathogens of plants.

This investigation aims for the wet chemistry-based synthesis of ZnNPs through the use of different (three each) reducing and complexing/capping agents. The generated ZnNPs were characterized through UV-Vis spectroscopy, transmission electron microscopy (TEM), X-ray diffraction spectroscopy (XRD), and Fourier transform Infrared Spectroscopy (FT-IRS). These ZnNPs were evaluated for anti-fungal potential against three prominent fungal pathogens of maize *viz.*, *Curvularia lunata, Fusarium oxysporum,* and *Macrophomina phaseolina* in an agar well diffusion assay. The variation in the anti-oxidant potential of these ZnNPs was assessed through scavenging of the DPPH radicals while the genomic DNA degradation potential was determined through a DNA-ZnNPs incubation study followed by agarose gel electrophoresis of the samples.

## 2. Materials and Methods

### 2.1. Chemicals and Microbial Cultures

The zinc precursor salts (zinc acetate dihydrate (Zn(OAc)_2_·2H_2_O), zinc nitrate hexahydrate (Zn(NO_3_)_2_·6H_2_O), and zinc chloride (ZnCl_2_)) utilized in the study were analytical grade and purchased from HiMedia (HiMedia Laboratories, Mumbai, India). The reducing agents (sodium hydroxide (NaOH), sodium borohydride (NaBH_4_)) and other chemicals (ammonium hydroxide or ammonia solution (NH_4_OH, 25%), and thiourea (NH_2_CSNH_2_)) were purchased from Sigma-Aldrich (St. Louis, MO, USA). The analytical grade complexing-capping agents (starch, cellulose, and bovine serum albumin) were procured from Himedia (Himedia Laboratories, Mumbai, India). The HPLC grade water (CAS No. 7732-18-5, Sisco Research Laboratories Pvt. Ltd., Mumbai, India) was used for the preparation of all the solutions and broth and agar-based media. The broth (CAS No. M403) and agar-based (CAS No. MH096) potato dextrose media were purchased from Himedia (Himedia Laboratories, Mumbai, India).

The maize crop-specific three major test fungal cultures, *Curvularia lunata* (ITCC 7170)*, Fusarium oxysporum* (ITCC 7093), and *Macrophomina phaseolina* (ITCC 5467) were purchased from Indian Type Culture Collection, Division of Plant Pathology, Indian Agricultural Research Institute, New Delhi, India. The cultures were subcultured on Potato Dextrose Agar media and incubated at 25 °C in the dark.

### 2.2. Synthesis and Characterization of NMs

The ZnNMs were synthesized using two different approaches: Wet chemical and Sol-gel synthesis methods. Six different ZnNMs samples were prepared using soluble zinc salts (zinc acetate/zinc nitrate/zinc chloride) as the precursors. The reducing/complexing-capping agents used in the study were sodium hydroxide [25], thiourea [26] and natural polymers such as starch [27,28], and cellulose-nanocomposite [29], and protein (Bovine-serum albumin (BSA)) [30]. The brief protocols depicting the schematic steps have been provided in Figure 1.

#### 2.2.1. UV-Vis Spectroscopy

The spectral absorbance behavior of the synthesized nanoparticles was analyzed on a Double Beam UV-Vis Spectrophotometer (model Elico SL 218, India) through screening over wavelengths ranging from 190 to 800 nm.

#### 2.2.2. Transmission Electron Microscopy

The morphology of the synthesized nanoparticles was determined by the transmission electron microscopy (Hitachi H-7650, Japan) analysis. The powdered ZnNMs were ground in a polystyrene pestle mortar, homogenized, and suspended in a known volume of deionized water. The suspension was bath sonicated for 15 to 30 min and 20 μL of the suspension was then placed on a copper grid (carbon film-coated, 200-mesh size). The prepared grids were air-dried and viewed under high-resolution imaging mode in TEM operated at 80 kV acceleration voltage.

#### 2.2.3. X-ray Diffraction Spectroscopy

The crystal structure and size of the synthesized ZnNMs were obtained through X-ray diffraction spectroscopy analysis. The X-ray diffraction patterns for the six different ZnNMs were obtained on X-ray diffractometer (Bruker D8 Advance, Germany) using Cu (Kα λ = 0.150595 nm) radiations at specific operation conditions (voltage: 40 kV, current: 30 mA). The samples were scanned with a scanning angle (2θ) range of 5° to 80° and a step size of 0.02° respectively.

#### 2.2.4. Fourier Transform Infra-Red Spectroscopy

The FT-IR spectroscopy (Thermo Nicolet 6700, USA) equipped with DTGS (deuterated triglycine sulfate) detector and KBr beam splitter system was used to characterize and identify the functional group diversity of the prepared nanoparticles. The summation spectra were generated in transmittance mode by placing dry, homogenized ZnNMs powdered samples on zinc selenide (flatbed configuration) crystal of the Attenuated total reflection (ATR) (Smart assembly, Thermo Fischer, USA) assembly with operational parameters of 32 scans at 4.0 cm^−1^ spectral resolution and acquired for the mid-IR region spanning over 4000 to 600 cm^−1^.

### 2.3. Anti-Oxidant Activity of the ZnNMs

The anti-oxidant potential of the ZnNMs was determined as the free radical scavenging potential through neutralization of the 1, 1-Diphenyl-2-picrylhydrazyl (DPPH) radicals [31,32]. The aqueous suspensions of the ZnNMs were prepared by dispensing a known amount of the ZnNMs in a defined volume of HPLC-grade water. The prepared dispersions (1 mL) were bath sonicated, incubated under the same conditions with 3 mL methanolic solution of DPPH (0.1 mM) for half an hour. Variable color development in the incubated solutions indicating the radical scavenging rate was measured as absorbance at 517 nm. The percent inhibition was compared with the values obtained for the butylated hydroxytoluene (BHT) as standard.

### 2.4. Anti-Mycotic Activity of the ZnNPs

#### 2.4.1. Agar Well Diffusion Assay

The anti-fungal activity of the ZnNPs was evaluated through agar well diffusion assay [33] involving estimation of the mycelial growth-inhibiting potential of the synthesized ZnNPs on the three test maize pathogenic fungal cultures [31]. The PDA media was poured in sterilized petri dishes (90 × 14 mm, Tarsons triple vent radiation sterile polystyrene, Code: 460091, Tarsons, Kolkata, India) and allowed to gel. The wells in the solidified media were prepared using a sterile cork borer (diameter 8.0 ± 0.2 mm, CAS No. LA737, Himedia Laboratories, Mumbai, India). The ZnNPs stock solutions were prepared in the HPLC grade water and these suspensions were bath sonicated for 30 min at room temperature. The stock solutions were then utilized for the preparation of the working concentrations (0, 5, 10, 20, and 40 mg L^−1^). These suspensions were then given a quick bath sonication treatment for another five minutes and 20 μL of the suspensions were loaded in the agar wells. The respective fungal growth on PDA media served as the negative control. The fungal disc (8.0 mm diameter) of the freshly grown confluent growth (one-week old culture plate) was placed at the center of each plate and the inoculated petri plates were incubated in a BOD incubator at 27 ± 2 °C for seven days. The diameter of the zone of inhibition of the mycelial growth at or near the well containing the ZnNPs was taken as indicator of the decreased mycelial growth.

#### 2.4.2. Optical Research Microscopy Studies of the Fungal Hyphae

For the optical microscopy (Leica DM5000 B, Leica Microsystems, Germany) studies, fresh slides were prepared to observe the effect of ZnNPs on the hyphal morphology and structure representing the morphological damage caused to the fungi by the ZnNPs. The slides were stained with lactophenol-cotton blue dye to visualize the variations in the hyphal morphology at 200 and 400× magnifications and imaged (Leica DFC 420C, Germany).

### 2.5. Fungal Genomic DNA Degrading Potential of the ZnNPs

The fungal biomass of the three fungi was generated in potato dextrose broth. The fungal mats were filtered and washed through sterilized filter paper (Whatman qualitative filter paper No. 1, Sigma-Aldrich, USA). The mycelial mat was then placed in a ceramic pestle containing liquid nitrogen and finely ground to obtain a powder. The fungal genomic DNA was extracted [34], quantified for quality and quantity and known quantity (10 µL) was incubated with 40 µg mL^−1^ of ZnNPs (NaOH as reducing agent) for 2 and 24 h at 37 °C. The incubated genomic DNA was resolved on 1.5% (*w/v*) agarose gel containing ethidium bromide (0.05 µg mL^−1^). The resolved gel was viewed in a Gel Documentation and Analysis System (Uvitec, Cambridge, UK), and images were captured.

### 2.6. Statistical Analyses

The antioxidant profile data obtained for five replications were subjected to analysis of variance (ANOVA) by using the generalized linear model (Proc GLM) command for a completely randomized experimental design and results were obtained on analysis using SAS software (version 9.2, Cary, NC, USA). The mean comparisons were performed with the least significant difference (LSD, *p* ≤ 0.05) approximation.

## 3. Results

### 3.1. Characterization of ZnNPs

#### 3.1.1. UV-Visible Spectroscopy

Among the primary spectroscopy techniques utilized for the characterization of nanoparticles, this absorption spectroscopy technique is used to evaluate the light-matter interactions and has profound relevance for the determination of the optical properties of nanoparticles including key characteristics such as shape, size, and stability [35,36]. The UV-visible absorption study of the six different ZnNPs illustrated distinct and sharp absorption peaks to vary between wavelengths ranging from 210 to 350 nm (Figure 2). All the reducing and capping agents derived ZnNPs exhibited a single and sharp peak at 210 or around 210 to 220 nm except Starch (RA) (dual distinct peaks at 212 and 350 nm) and Starch (CA) (triple peaks at 212, 260, and 330 nm) derived ZnNPs.

#### 3.1.2. Transmission Electron Microscopy

The TEM micrographs exhibit the occurrence of spherical to hexagonal-shaped ZnNPs having well-defined crystal edges and planes (Figure 3a–f). Partial (Figure 3c,e,f) to high (a, b, and d) agglomeration can also be observed. The nanoparticle size distribution was substantially variable for the reducing/capping agents used for the generation of ZnNPs. The lowest size distribution ranges of 8 to 26 nm and 6 to 22 nm were observed for thiourea (RA) and starch (CA) ZnNPs respectively. However, the starch (CA) ZnNPs appeared to be adorned on electronically less dense substrate material possibly derived from burning/charring of the starch during the calcination process. The average mean diameter of the nanoparticles (nm ± S.E.) was as follows; NaOH (RA) (31.37 ± 2.48), thiourea (RA) (15.86 ± 0.59), starch (RA) (36.82 ± 2.41), albumin (CA) (82.94 ± 3.64), starch (CA) (11.78 ± 0.74) and cellulose (CA) (209.18 ± 15.02).

#### 3.1.3. X-ray Diffraction Spectroscopy

The XRD patterns varied in peak intensity and width according to the reducing/capping agent used. In general, the peak widening could be observed for all the six types of synthesized ZnNPs which represents the nano-scale crystalline size of the prepared ZnNPs. The XRD peaks obtained showed close agreement with the characteristic Bragg peaks of hexagonal ZnO zincite (pattern: PDF 00-036-1451), and hydro-zincite (pattern: PDF 01-072-1100). Mixed crystal phases including the simonkolleite (pattern: COD 9004683) along with zincite and hydro-zincite specific peaks can be identified in ZnNMs derived from ZnCl_2_ salt as substrate (Figure 4). The use of starch as a reducing agent resulted in the formation of nanoscale hydrozincite crystals. While Starch (CA) ZnNPs had both zincite and hydrozincite as the predominant crystal phases. This variation may be attributed to the use of NaOH as a reducing agent during the synthesis of the Starch (CA) ZnNPs. Cellulose (CA) ZnNPs possessed a mixed crystal phase with simonkolleite as one among the predominant crystal structure. As indicated above, it can probably be due to the use of a high concentration of zinc chloride (65 wt %) as the precursor salt for the synthesis of ZnNPs.

#### 3.1.4. Fourier Transform Infra-Red Spectroscopy

The characteristic FTIR peaks for metal oxides appear in the fingerprint region of 1700 to 600 cm^−1^ due to vibrations among the metal and other atoms (O or OH) associated with it [37]. The NaOH (RA) ZnNPs exhibited specific peaks in this region at 663.0, 815.2, 952.0, 1016 (ν_1_ frequency), and 1624 cm^−1^ featuring Zn-O bond deformation and stretching vibrations respectively [37]. The presence of adsorbed water molecules can be identified due to the appearance of a broad peak at 3443.0 and a sharp peak at 1103.0 cm^−1^ which can be ascribed to stretching and deformation vibrations of the O-H bond (Figure 5). Further, the occurrence of the hydrozincite phase in the starch (RA), albumin (CA), and starch (CA) ZnNPs, the stretching vibrations of the C=O and C–O bonds in the carbonate functional group (CO_3_^2−^) can be ascribed to bands at 1360 and 1407 cm^−1^. The conspicuous broadband (3690 to 2975 cm^−1^) was observed in starch (RA), and albumin, starch and cellulose (CA) which can be ascribed to hydroxyl (O–H) and amine (N–H) group vibrations. Similar peaks have also been reported for thiourea derived ZnNPs [26].

### 3.2. Anti-Oxidant Activity of the ZnNPs

The DPPH synthetic radicals are considered relatively stable to evaluate the radical scavenging potential of nanoparticles or other compounds. The prepared ZnNPs exhibited significant free radical scavenging activity (DPPH FRSA) which ranged from 25 to 84% inhibition (Figure 6). The NaOH (RA) ZnNPs possessed the highest antioxidant activity. The order of the FRSA was NaOH (RA) > Starch (RA)/(CA) > Albumin (CA) > Cellulose (CA) > Thiourea (RA).

### 3.3. Anti-Mycotic Activity of the ZnNPs

#### 3.3.1. Agar Well Diffusion Assay

Post three days of incubation, the prepared ZnNPs were evaluated for five different concentrations (0, 5, 10, 20, and 40 mg mL^−1^) to exhibit an inhibitory effect on the mycelial growth for all three test fungi. Among the three fungal cultures, maximum mycelial inhibitory activity was recorded in order *Fusarium oxysporum* > *Curvularia lunata* > *Macrophomina phaseolina* (Figure 7). The efficacy of the NaOH (RA) derived ZnNPs was identified by the formation of a larger inhibition zone compared to the other ZnNPs evaluated. The control well containing only sterilized HPLC grade water did not exhibit any antifungal activity (Figure 7). The radial diameter of all the three-test fungi was the smallest for the NaOH (RA) derived ZnNPs particularly clear cottony-white hyphal inhibition could be observed for *Macrophomina phaseolina* (Figure 7d). Moreover, the sparse, aerial, and fluffy fungal growth of the *Fusarium oxysporum* indicates the response of hyphae to ZnNPs stress. This is the first report on variability in the anti-mycotic efficacy of the ZnNPs derived from different reducing/complexing agents on maize crop-specific pathogenic fungi.

#### 3.3.2. Optical Research Microscopy Studies of the Fungal Hyphae

The optical research microscopy of the mycelial growth at the fringes of the colony exhibited variation in the hyphal morphology as observed through appearance of swelling/rolling, thinning, fragmentation, and hyper-branching of the mycelia. The hyphae of all the three fungal genera sampled near the well containing NaOH (RA) (40 mg L^−1^ concentration) showed cell wall distortion, cytoplasmic shrinkage, oozing out of the cytoplasmic material, and hyphal fragmentation. The optical micrographs of the *Fusarium oxysporum* hyphae in NaOH (RA), Albumin (CA), and Starch (CA) ZnNPs (40 mg L^−1^ concentration) treatment showed extensive leakage of the cytoplasmic material from the hyphal tissue (Figure 8). The swelling of the hyphae cells can also be identified in the Thiourea (RA) ZnNPs and NaOH (RA) ZnNPs for *Curvularia lunata*, and *Macrophomina phaseolina* respectively.

### 3.4. Fungal Genomic DNA Degrading Potential of the ZnNPs

Incubation of the genomic fungal DNA with NaOH (RA)-derived ZnNPs resulted in fragmentation of the DNA which can be identified as smeared DNA appearance in the 1.5% agarose gel compared to the intact DNA band in the control lane of the three test fungi (Figure 9). After 2 h of incubation with the NaOH (RA) derived ZnNPs (40 μg mL^−1^ concentration), a slight decrease can be noticed in the genomic DNA of all the test fungi in lanes 4, 5, and 6 (Figure 9). Therefore, the incubation was extended until 24 h to observe any further impact on the genomic DNA of these fungal genera. A clear fragmentation and decrease in the genomic DNA content can be visualized in lanes 7, 8, and 9 as compared to lanes 1, 2, and 3 respectively (Figure 9).

## 4. Discussion

The UV-Vis absorption peaks exhibited all the ZnNPs are way below the characteristic excitonic absorption peak value of 370 nm attributed to intrinsic band-gap absorption phenomena exhibited by the bulk ZnO under valence to conduction band electron transitions [36,38]. Similar reports of blue-shifted UV vis absorption peaks have been reported for ZnNPs synthesized through sol-gel [25,39], microemulsion route [37], and green synthesis approach [40]. Moreover, a similar UV-Vis peak at 347 nm has been observed for the starch (CA)-ZnNPs on laser ablation synthesis in an aqueous starch solution for 15 to 20 min which could be ascribed to the formation of a layered nanocomposite comprised of starch-β-Zn(OH)_2_ sheets [41]. The occurrence of a single sharp peak indicates the formation of monodispersed ZnNPs and therefore, the narrow size distribution pattern of the synthesized ZnNPs [42]. The dual or triple peaks indicate the polydisperse nature of the aqueous nano-sol probably due to the formation of larger agglomerates by coalescence during or post-nucleation process [43]. The albumin (CA) ZnNMs showed a sharp peak at 220 nm, and multiple stout peaks at 400, 620, 700, and 760 nm wavelength which showcased the highly polydisperse nature of the albumin (CA) ZnNMs aqueous sol. However, the small area under these multiple peaks is also indicative of the presence of variable larger-sized ZnNPs in low amounts.

On the estimation of the bandgap energy of different ZnNMs according to Einstein equation; Energy = hC/λ, where h = Plank’s constant (6.626 × 10^−34^ Joules sec), C = Velocity of light (3 × 10^8^ m s^−1^) and λ = wavelength (nm), the calculated values were 5.90, 5.64, 3.76 and 3.50 eV for the sharp and distinct absorption peak wavelengths of 210, 220, 330 and 350 nm respectively. These values are quite high compared to the moderate band-gap energy (3.35 eV) of ZnO which corresponds to UV and deep blue region indicating O_2p_ to Zn_3d_ electron transitions [44]. These blue shifts in the band-gap energy may be ascribed to the inadvertent occurrence of other metal atom impurities that may have altered the electronic movement across the valence and conduction bands [45]. Similar blueshifts have been observed on doping of zinc with IIIa group metal atoms [46] and 3d transitions metals [45].

The particle size of less than 10 nm have been reported for NaOH-ethanol reaction mix [43] while ZnNPs generated from NaOH-isopropanol reaction mixture formed particles with the mean size dimensions of 189.0 ± 6.0 and 447.0 ± 22.0 nm by sonochemical and hydrothermal techniques respectively [47]. However, unlike Chen et al. [30] the ZnNPs size was larger for the albumin (CA) ZnNPs while similar size distribution and average size have been observed for thiourea (RA) [26], cellulose (CA) [29], and starch (CA) ZnNPs [27].

The use of a high concentration of ZnCl_2_ can lead to the formation of hexagonal layered plate-like crystals of simonkolleite [18]. However, the XRD spectra for NaOH and thiourea reducing agents exhibited the presence of only hexagonal zincite phase crystals. Similar diffraction peaks have been reported for ZnNPs obtained by green synthesis from fruit parts of *Citrullus colosynthis* [14] and cotton linter pulp [11]. The XRD spectra of the albumin (CA) ZnNMs showed a very broad peak spanning over 30 to 50 20° diffraction region with several substantially indistinct peaks within this range indicating the occurrence of mixed crystal phase. As Bovine serum albumin (BSA) exhibits affinity to adsorb or form protein corona structure on the surface of the ZnNPs [19,20], it may have restrained the coalescence of the smaller-sized particles to form larger aggregates during the nucleation process. However, the transmission electron micrograph indicated the formation of plate-like hexagonal ZnNPs (Figure 2).

The DPPH FRSA of the ZnNMs may be ascribed to the transfer of electron density from oxygen atom in ZnNMs to N-atom odd electron in DPPH compound [48]. The antioxidant properties further depict the efficiency of the redox-catalysis reactions, electronic configuration, surface-interface effect, and biocompatibility of the ZnNMs. From the results obtained for the FRSA potential, it appears that the NPs of a particular size is critical beyond which any further reduction in the size of the NPs does not contribute significantly to the antioxidant behavior. These observations are in line with the anticipated size-dependent phenomena and larger surface area to volume ratio to be involved in improved neutralization/deactivation of the hydroxy (^•^OH) radicals [49]. Similar to observations of this study, the report on electron spin resonance (ESR) spectroscopy study of 4 to 15 nm AuNPs, showcased maximum antioxidant activity by 9 nm NPs and not by NPs with a size smaller than 9 nm [49]. Therefore, it can be argued that the dose [50,51] or concentration [52], specific surface, and crystallinity status [49] of the ZnNMs are relatively more critical features for altering the efficiency with which the ZnNMs interact with the DPPH radicals.

The fungal radial growth inhibition in the agar-well diffusion assay for the three test fungi indicated superior antimycotic activity of NaOH (RA) derived ZnNPs. Similar radial growth inhibition has also been reported for ZnO NPs by He et al. [11] for *Penicillium expansum* and *Botrytis cinerea*. The hyphal thinning effects can be observed for most of the ZnNMs evaluated in the study which appears to be a characteristic feature of any fungal tissue in response to nanoparticle challenge [53,54]. However, the morphological manifestations such as breakage of the cell wall and leakage of the cytoplasmic contents, and appearance of swollen hyphal cells indicate the alteration in the osmotic conditions and formation of physical pores in the cell wall/membrane of the treated fungal cells [19,23]. Coherent to the results of the mycelial inhibition in agar well assay as described in the previous Section 3.3.1, maximum impact on the hyphal morphology was recorded for the *Fusarium oxysporum*. However, the hyphal fragmentation can be observed in *Macrophomina phaseolina*, unlike the agar well assay which indicates the subtle cytological changes that occur in the fungal hyphae on treatment with ZnNMs. A substantially low concentrations of the ZnNMs have been evaluated in this study unlike the studies performed on *Pythyium* [23], *F. graminearum, A. flavus*, and *P. citrinum* [19], and *Aspergillus flavus* [14].

The genomic DNA fragmentation may be attributed to the physical and chemical properties of the NPs [55] including the size, concentration [56,57], chemistry [56,57,58], and surface functionalization [59]. Though within a fungal cell the predominant mechanism of degradation of the cellular DNA by ZnNPs is through formation of reactive oxygen species (ROS) which causes extensive DNA scissoring and fragmentation [7]. However, direct interactions of ZnNPs with DNA molecules involve the binding of DNA with ZnO nanoparticles to nucleobases [60]. The specific conjugation of the DNA on ZnO NPs surface has been further identified by Das et al. [61] evidenced through varied conductivity and mobility under electric field on an agarose gel. The results of this study effectively demonstrate that the ZnNPs can intercalate with the DNA and exhibit non-photocatalytic DNA degradation unlike the sunlight-induced fragmentation of *Leishmania* DNA on incubation with Ag-doped ZnO NPs [62].

## 5. Conclusions

This study provides evidence on variation in the anti-mycotic, anti-oxidant, and DNA degrading effect of the ZnO nanoparticles generated through wet chemical synthesis methods by using different reducing or complexing/capping agents. Both the anti-oxidant and anti-mycotic potentials were not observed to follow strict nanoparticle size-dependence. Thus, the NaOH (RA) derived ZnNPs had zincite crystal phase and the particle size distribution ranging from 10 to 90 nm and possessed the highest antioxidant and antimycotic activities. This study also illustrates the subtle hyphal morphological changes occurring on the use of lower concentration of the ZnNPs which may span over a variety of manifestations ranging from thinning, fragmentation, swelling, and lysis of the fungal hyphae. These alterations in the hyphal morphology could be attributed to variability in the functionalizations or chemical functional groups present on the surface of the ZnNPs. Further, the fungal DNA-NaOH (RA) ZnNPs incubation study enunciated the DNA degradation and exacerbation of the total genomic DNA of the test pathogenic fungi in a concentration and time-dependent manner with complete degradation of the *Macrophomina phaseolina* genomic DNA after 24 h of incubation. It could also be inferred from the study that the DNA damaging effect of the ZnNPs is also fungal genera/species-specific and may vary according to the fungal culture being studied. Therefore, this work establishes the impact of the appropriate nano-crystallite size dimensions and surface functionalizations as the key factors that vary the anti-mycotic, anti-oxidant, and DNA degrading potentials of ZnNPs.

## Figures and Tables

**Figure 1 jof-07-00223-f001:**
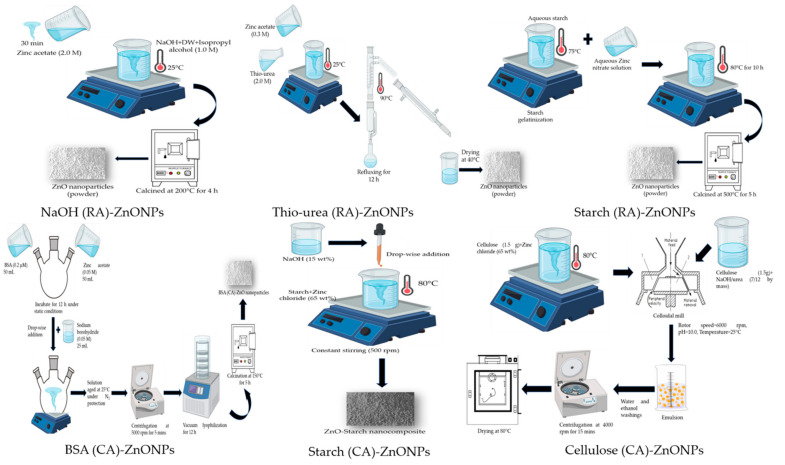
Schematic representation of steps for the synthesis of zinc nanomaterials using different reducing and complexing/capping agents.

**Figure 2 jof-07-00223-f002:**
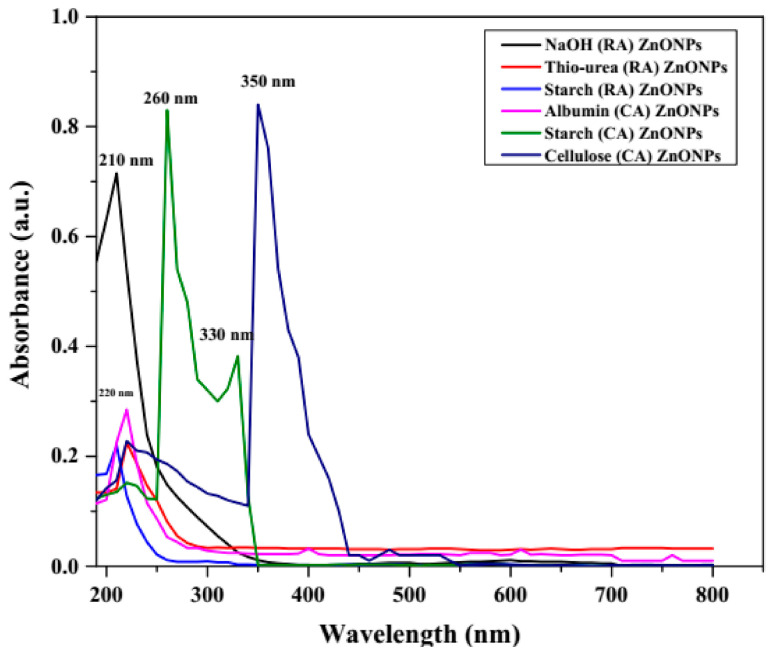
Variable UV-Vis absorbance spectra of the synthesized ZnO nanoparticles. RA: reducing agent, CA: capping/complexing agent.

**Figure 3 jof-07-00223-f003:**
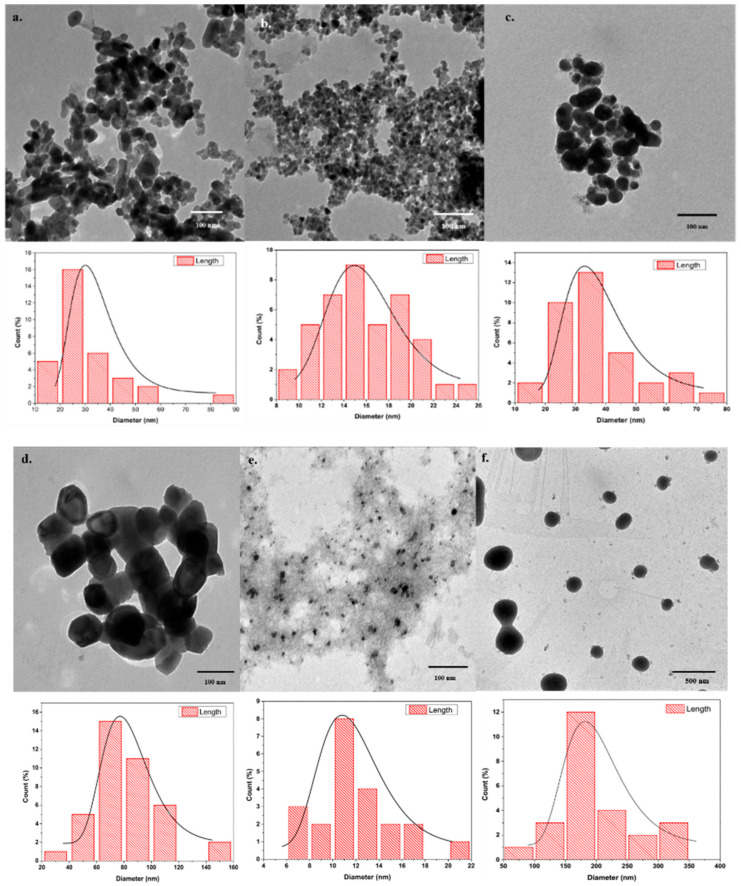
Transmission electron micrographs depicting the variation in the ZnO nanoparticle dimensions for the reducing (RA) and capping/complexing (CA) agents. (**a**) Sodium hydroxide (RA), (**b**) Thiourea (RA), (**c**) Starch (RA), (**d**) Bovine serum albumin (CA), (**e**) Starch (CA), and (**f**) Cellulose (CA).

**Figure 4 jof-07-00223-f004:**
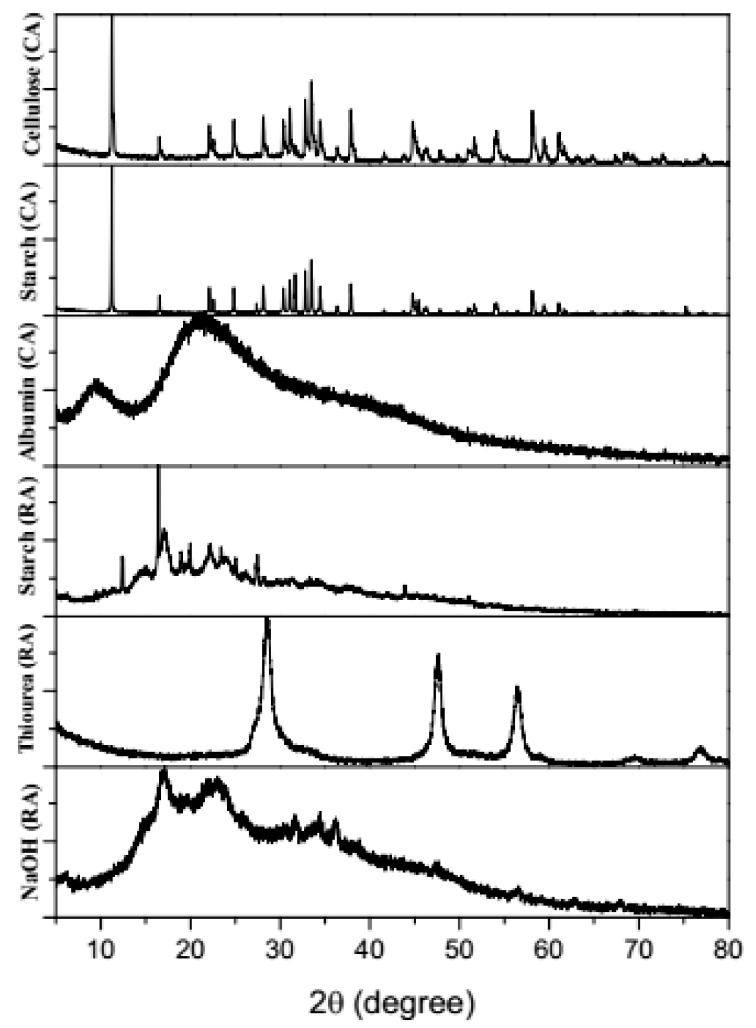
X-ray diffraction spectroscopy of the synthesized ZnNPs depicting the formation of wurtzite hexagonal crystal structure zincite (ZnO) on the use of various reducing and capping/complexing agents i.e., Sodium hydroxide (RA), Thiourea (RA), Starch (RA), Bovine serum albumin (CA), Starch (CA), and Cellulose (CA).

**Figure 5 jof-07-00223-f005:**
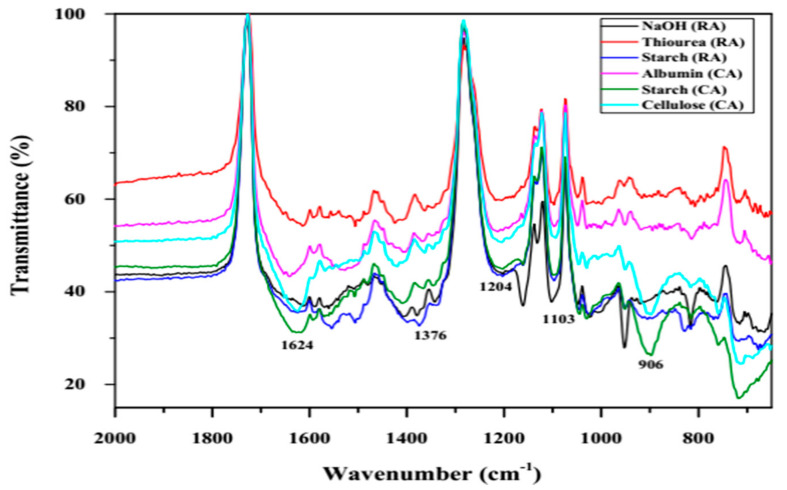
FT-IR cumulative spectra of the prepared ZnNPs for the mid-IR region (2000 to 650 cm^−1^ wavenumbers) indicating variability in the occurrence of chemical functional groups.

**Figure 6 jof-07-00223-f006:**
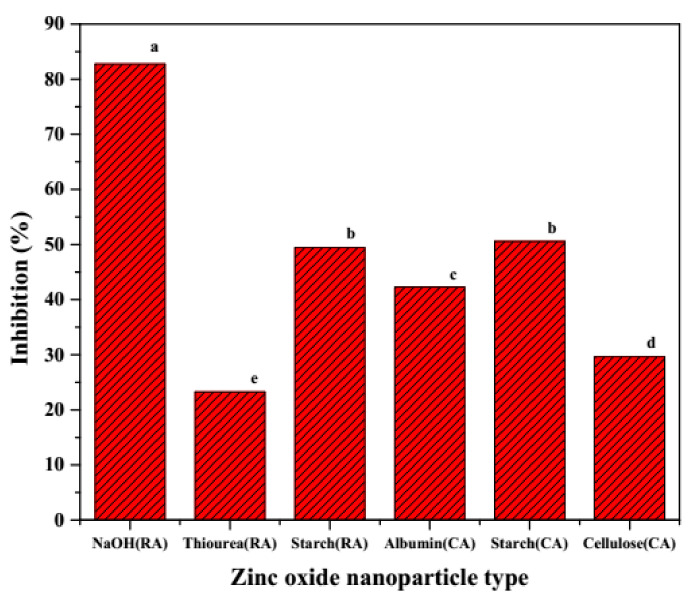
Comparative antioxidant potential of ZnNPs as determined through scavenging activity (%) of DPPH radicals. Different letters denote a significant difference (*p* ≤ 0.05) among six different types of ZnNPs.

**Figure 7 jof-07-00223-f007:**
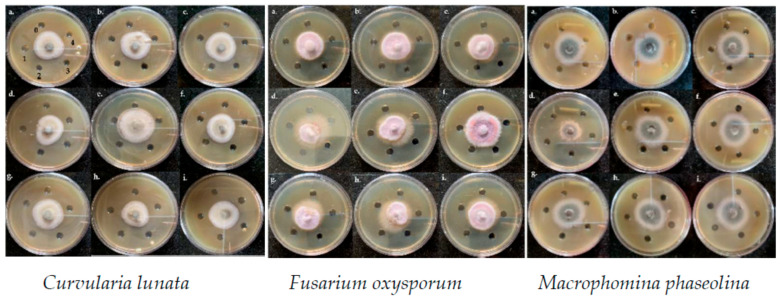
Effect of different ZnNPs and zinc salts on hyphal growth of three maize pathogenic cultures, Curvularia lunata, Fusarium oxysporum, and Macrophomina phaseolina. (**a**) Zinc acetate, (**b**) Zinc chloride, (**c**) Zinc sulphate, (**d**) Sodium hydroxide (RA) ZnNPs, (**e**) Thiourea (RA) ZnNPs, (**f**) Starch (RA) ZnNPs, (**g**) Bovine serum albumin (CA) ZnNPs, (**h**) Starch (CA) ZnNPs, and (**i**) Cellulose (CA) ZnNPs. The figures from 0 to 4 indicate different concentrations of the zinc salts and ZnNPs. 0 = distill water, 1 = 5 mg L^−1^, 2 = 10 mg L^−1^, 3 = 20 mg L^−1^, 4 = 40 mg L^−1.^

**Figure 8 jof-07-00223-f008:**
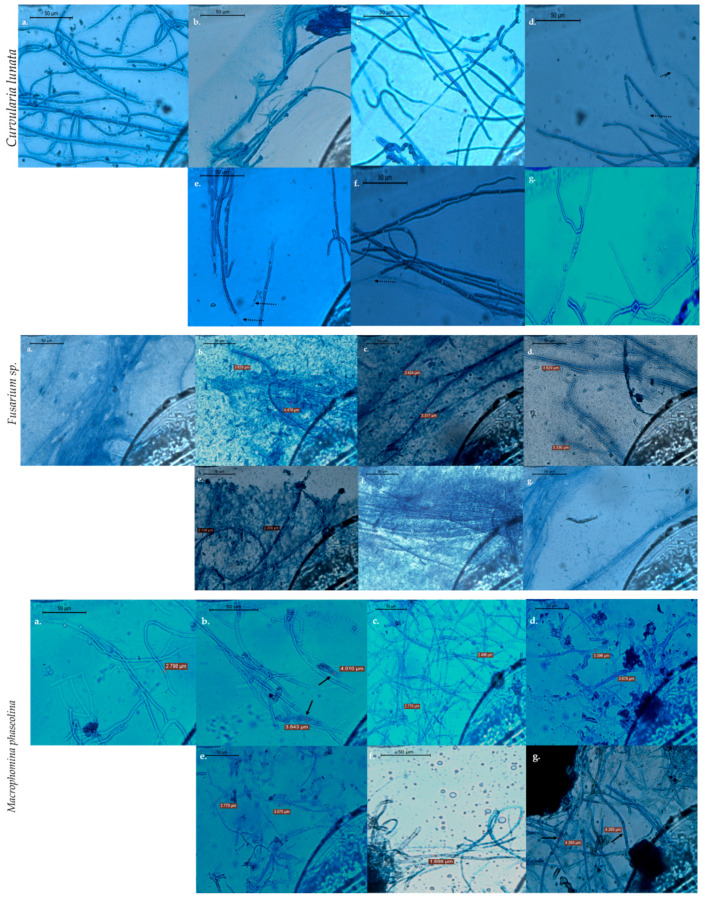
Optical micrographs of the three test fungal cultures depicting cytological events such as hyphal fragmentation, clearing of the cell cytoplasm, hyphal thinning, and dissolution of the fungal cell wall on incubation with ZnO nanoparticles derived from various reducing and capping/complexing agents. (**a**) Control, (**b**) Sodium hydroxide (RA), (**c**) Thiourea (RA), (**d**) Starch (RA), (**e**) Bovine serum albumin (CA), (**f**) Starch (CA), and (**g**) Cellulose (CA). Magnification-400×. The solid arrow indicates the thickening of the hyphae, dotted arrow indicates the clearing of cell cytoplasm.

**Figure 9 jof-07-00223-f009:**
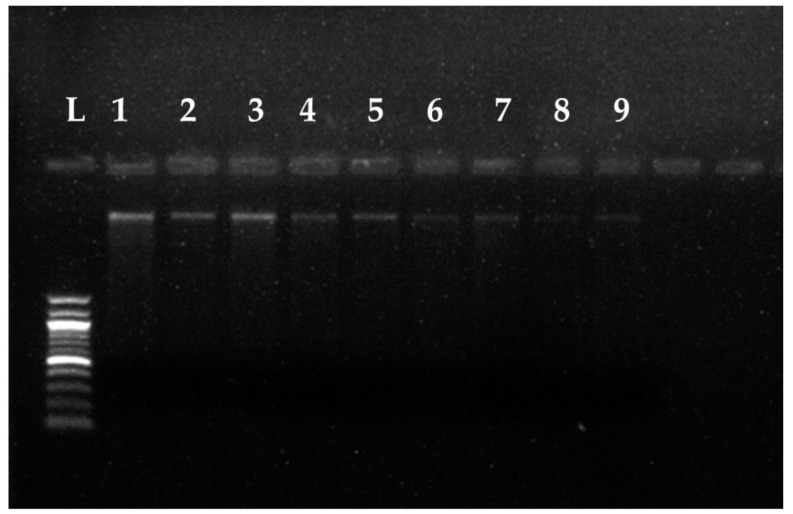
Fragmentation and degradation of the fungal genomic DNA on incubation with NaOH (RA)-derived ZnNPs. Lane L = Marker ladder, Lane 1 = gDNA of *Fusarium oxysporum* (*FO*), Lane 2 = gDNA of *Curvularia lunata* (*CL*), Lane 3 = gDNA of *Macrophomina phaseolina* (*MP*), Lane 4 = *FO* gDNA incubated with ZnNPs for 2 h, Lane 5 = *CL* gDNA incubated with ZnNPs for 2 h, Lane 6 = *MP* gDNA incubated with ZnNPs for 2 h, Lane 7 = *FO* gDNA incubated with ZnNPs for 24 h, Lane 8 = *CL* gDNA incubated with ZnNPs for 24 h, Lane 9 = *MP* gDNA incubated with ZnNPs for 24 h.

## Data Availability

Data is contained within the article.

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
