# Peer review of "Differential Antimycotic and Antioxidant Potentials of Chemically Synthesized Zinc-Based Nanoparticles Derived from Different Reducing/Complexing Agents against Pathogenic Fungi of Maize Crop"

_jof, 2021, doi:10.3390/jof7030223_

Round 1
Reviewer 1 Report
After checking the changes authors have made, after my previous revision, I am pleased to recommend the revised manuscript for publication in its current form.
Reviewer 2 Report
Authors have made suitable improvement in the manuscript. It may now be accepted in JOF
This manuscript is a resubmission of an earlier submission. The following is a list of the peer review reports and author responses from that submission.
Round 1
Reviewer 1 Report
This paper describes the different antimycotic and antioxidant acrivity of ZnO particles synthesized with different methods It has originality and overall presentation is good. However, there are some unclear points
Main concerns:
1 the abstract is too long, please summarize and better highlight the objectives of the work
2 please write the synthetic procedures of the various synthesized materials
3 in figure 2.f please improve the size statistics
4 the figures 4 and 5 are not clear, please highlight the peaks better
5 explain better how the percentage of inhibition shown in Figure 6 was calculated
6 better describe the correlation between ZnO particle size with antioxidant activity, in particular explain better from line 313 to line 321
7 in Figure 8 better highlight the differences between the control and the various treatments with ZnO
Reviewer 2 Report
The experimental details especially for the synthesis of materials are unclear.
One does not know which chemicals were used as reductants and which were used as capping material.
Most of the chemicals used will not result in the formation of ZnO as described by the authors. Interaction of zinc salts with NaOH and NH4OH will result in hydroxide forms of Zn. Similarly other chemicals like urea might result in a complex formation. Therefore the term ZnO is highly inappropriate. The authors have no idea of the material they are studying.
Most of the samples are not nanoparticles as claimed by authors. For example material produced by interaction with bovine serum albumin and cellulose are too large to be designated general nanoparticles blanket.
The biological results maybe directly affected by the resulting material which is not ZnO in most cases.
The results maybe interesting but are scientifically unsound and therefore I can not recommend publication of this manuscript in JOF.